# Optimal Modeling of Sustainable Land Use Planning under Uncertain at a Watershed Level: Interval Stochastic Fuzzy Linear Programming with Chance Constraints

**Bingkui Qiu [1], Yan Tu [2], Guoliang Ou [3,\*], Min Zhou [4], Yifan Zhu [4], Shuhan Liu [4] and Haoyang Ma [4]**

[1] Department of Tourism Management, Jin Zhong University, Jinzhong 033619, China; qbk@jzxy.edu.cn
[2] Hunan Institute of Science and Technology Information, Changsha 410001, China; 18674861067@163.com
[3] School of Construction and Environmental Engineering, Shenzhen Polytechnic, Number 7098, Liuxian Dadao, Nanshan District, Shenzhen 518055, China
[4] College of Public Administration, Huazhong University of Science and Technology, Wuhan 430074, China; m202275029@hust.edu.cn (Y.Z.); liu_shuhan@163.com (S.L.)
[\*] Correspondence: ouyang8305@szpt.edu.cn; Tel.: +86-755-26731157; Fax: +86-755-26731436

**Abstract:** In this paper, an uncertain interval stochastic fuzzy chance constraint land use optimal allocation method is proposed and applied to solve the problem of land use planning in river basins. The UISFCL-LUP method is an aggregation of interval parametric programming, fuzzy linear programming and chance constraint programming which can cope with uncertain problems such as interval value, fuzzy set and probability. In this paper, the uncertain mathematical method is explored and studied in the optimal allocation of land use in the next two planning periods of Nansihu Lake Basin in China. Moreover, it was proved that ISFCL-LUP can deal with the uncertainty of interval, membership function and probability representation and can also be used to solve the land use planning and land use strategy analysis under uncertain conditions. On the basis of model calculations, we obtained the optimal allocation results for six types of land use in four regions over two planning periods based on different environmental constraints. The results show that the optimized λ value (that is, the degree of satisfaction with all the model conditions) is in the range of [0.54, 0.79] and the corresponding system benefits are between [18.4, 20.4] × 1012 RMB and [96.7, 109.3] × 1012 RMB. The results indicate that land managers can make judgments based on the different socio-economic development needs of different regions and determine strategic land use allocation plans under uncertain conditions. At the same time, the model obtained interval solutions under different system satisfaction and constraint violation probabilities, which helps land managers to analyze the importance of land system optimization and sustainable development more deeply.

**Keywords:** land use planning; uncertainty; interval fuzzy chance model; environmental protection; Nansi Lake Basin





## 1. Introduction

In the process of global industrialization and urbanization, the scarcity of land resources is one of the important problems [1]. This situation can cause various ecological and environmental problems such as land degradation [2], soil erosion, water pollution [3,4], biodiversity loss [5,6] and so on [7–9]. At present and in the foreseeable future, land resources are indispensable to economic development in China and most developing countries [10]. In this situation, land managers have to balance the relationship between economic growth and ecological protection through land use planning [11]. Land use allocation is the key to land use planning. Consequently, optimal land use allocation is considered one of the most effective methods, which can make land use more comprehensive, scientific and sustainable. Land use allocation can be defined as the uniform arrangement, by land managers, of all types of land (including agricultural land, industrial land, green land,

water area, etc.) in quantity and space within a certain period, based on the requirements of national economic development [12]. Zhai et al. (2022) proposed that the territorial space development pattern of a target construction should be able to support full circulation and optimal allocation of social elements and resources; the society in the region should be relatively fair; development opportunities and people's welfare should be equal; and the development of people, society, economy and environment is coordinated and sustainable. However, the uncertainty of realistic conditions and the complexity of land use systems make it difficult to practice scientific and rational land use planning [13].

In order to solve the above-mentioned problems, experts have conducted in-depth studies. In recent years, mathematical model methods have been applied to solve the problems of land use allocation and have played an important role in quantitative land use analysis and simulation research. Based on the review and classification sorting of previous research literature, these models can be divided into the following types: the first is the simulation prediction model [14–20], the second is the programming model [21–27], the third is the spatial model [28–31] and the fourth is the intelligent model [32–35]. For example, Mi et al. (2015) used an ant colony algorithm combined with a genetic algorithm. The genetic ant colony algorithm was applied to the optimal allocate model, thereby improving the calculation efficiency and the optimal spatial allocation of land use in the study area was obtained [36]. Lv et al. (2018) proposed a mathematical model of land use planning, based on a Monte Carlo simulation, to obtain distribution functions of random parameters, which has been exploratively applied to the in sustainable development planning of the Zhuhai regional ecosystem [37]. Ma and Zhou (2018) established a programming model of urban land resources allocation based on geographical information system (GIS) to support the spatial allocation of land resources under various ecological, environmental and socio-economic conditions, which is outstanding in expressing uncertainty, coupling model and spatial analysis [24]. Aburas and Ahamad (2019) reviewed the research on modeling, simulation and prediction models of land use change and thought that a machine learning model could simulate all the driving factors of land use change and is a powerful simulation model of land use change [38]. Cao and Zhang (2019) used a multi-objective optimization model of spatio-temporal land use, which scientifically shows the optimal evolution track of land use change within the planning time range [39]. Huang and Song (2019), in order to reconstruct the local optimization ability of the model, explored the optimal allocation model of land use by combining the multi-agent system based on land use planning knowledge with the search iteration mechanism in the mixed shuffled frog-leaping algorithm [40]. Li et al. (2020) constructed a mixed model framework of nonlinear programming and multi-objective programming which objectively reflected the actual problems in the allocation of agricultural water resources and land resources and applied the model to the study of land use allocation in the grain production base in northeast China [41]. Oleron-Evans and Salhab (2021) used multi-objective linear programming to investigate whether these figures were achievable given constraints on land availability and land use mix. How land uses might best be assigned to maximize home, job and gross value-added (GVA) creation within the Heathrow Opportunity Area was also explored [42]. Ma et al. (2022) proposed a collaborative optimal allocation of urban land (COAUL) model which provides a simulation tool both for the quantity and spatial structure optimization of urban agglomeration [43].

In previous studies, scholars have actively explored and developed a variety of mathematical models in order to solve the problem of optimal allocation of land use quantitatively. These models have their own advantages in space optimization, simulation allocation, algorithm optimization and other aspects and have been successfully applied. Unfortunately, due to the uncertainty of realistic conditions and the complexity of land use systems, these models still have two major shortcomings. On the one hand, due to the limited implementation conditions and the high data requirement standards of models, most models will face technical problems such as data barriers in the application process, which causes a certain lack of accuracy in the model data. On the other hand, due to the complicated

actual situation and various land use systems, it is difficult for the above models to limit the uncertain complexity in the real world for quantitative calculation such as the uncertainty of the government's investment in the land use system and the instability of the labor force used for the operation of the land use system.

Fortunately, the above two problems can handled well by the uncertain mathematical model. The uncertain mathematical model has been actively explored in the study of optimal allocation of water resources, crops and other resources and has been successfully applied [44–51]. For example, Ren and Li (2019) constructed an improved stochastic fuzzy multi-objective programming method by which decision makers can make appropriate decisions on the optimal utilization of irrigation water and soil resources under multiple different objectives and uncertain complex conditions [52]. At the same time, in the field of land use allocation research, scholars explored the use of uncertain mathematical models and conducted empirical studies [53–58]. For example, Zhou (2015), based on a combination of interval parametric programming, fuzzy flexible linear programming and opportunity constraint planning technology, established an interval fuzzy opportunity constraint land use allocation model to study land use planning under uncertain conditions [59]. Gu et al. (2020) used an uncertain fractional joint probabilistic opportunity-constrained programming method to obtain a series of land use policies in multiple scenarios [58].

In existing studies, uncertain mathematical models have played a role in dealing with the realistic constraints and problems in the optimal allocation of land use. However, there is still some room for improvement in the application of the uncertain model. First of all, it is difficult to fully clarify and quantify the complex relationships among the elements of land use system in the model, such as economic factors, social factors, environmental factors, etc. In recent years, a number of experts have devoted themselves to research in this area. For example, Song and Zhang (2021) thought that cultivated land use layout adjustment (CLULA) based on crop planting suitability is the refinement and deepening of land use transformation, which is of great significance for optimizing the allocation of cultivated land resources and ensuring food security [60]. Zhao et al. (2019) proposed coordinating land and sea management to restore natural habitats that were expanded into the high ES area. Implementing each ES (ecosystem services) weight of optimal scenario in land use management contributed to achieving inter-coordination of ES [61]. Arjomandi et al. (2021) aimed to determine the optimal land use allocation according to the economic, social and environmental criteria in the Hablehroud watershed using the multi-criteria decision making (MCDM) method [12]. Elalamy et al. (2019) developed a bio-economic model linking land use and ecosystem services to investigate the role of forests in a wide range of ecosystem services, including carbon sequestration, soil quality and biodiversity [62]. Han and Li (2021), based on the multi-regional decomposition analysis, investigated the embodied agricultural land flows among 31 provinces/municipalities of China and classified the transfer patterns into different drivers including intensity-, trade- and specialization-driven types. Secondly, the urban and national scales are the hot research areas of land use allocation, while the watershed scale has not received enough attention [63].

Therefore, the main task of this study is to propose a watershed land use planning model that can balance economic development and ecological environment protection in complex land use systems. Based on the above tasks, this paper proposes a linear land use planning model with uncertain interval stochastic fuzzy chance constraints (UISFC-LUP). The research results can help land use managers and decision makers understand how to make tradeoffs between economic development and ecological environment protection under the constraints of realistic conditions and formulate more rational and scientific land use planning programs.

## 2. Study Area and Data Sources

### 2.1. Study Area

Nansi Lake (34°24′–35°59′ N, 115°02′–117°42′ E) is the collective name of four lakes in series: Nanyang Lake, Dushan Lake, Zhaoyang Lake and Weishan Lake, located in the southwestern province of Shandong (Figure 1). The study area of this paper covers all areas of the Nansi Lake Basin in Shandong Province, inclusive of Jining City, Zaozhuang City, Heze City and Ningyang County, that is, three cities and one county.

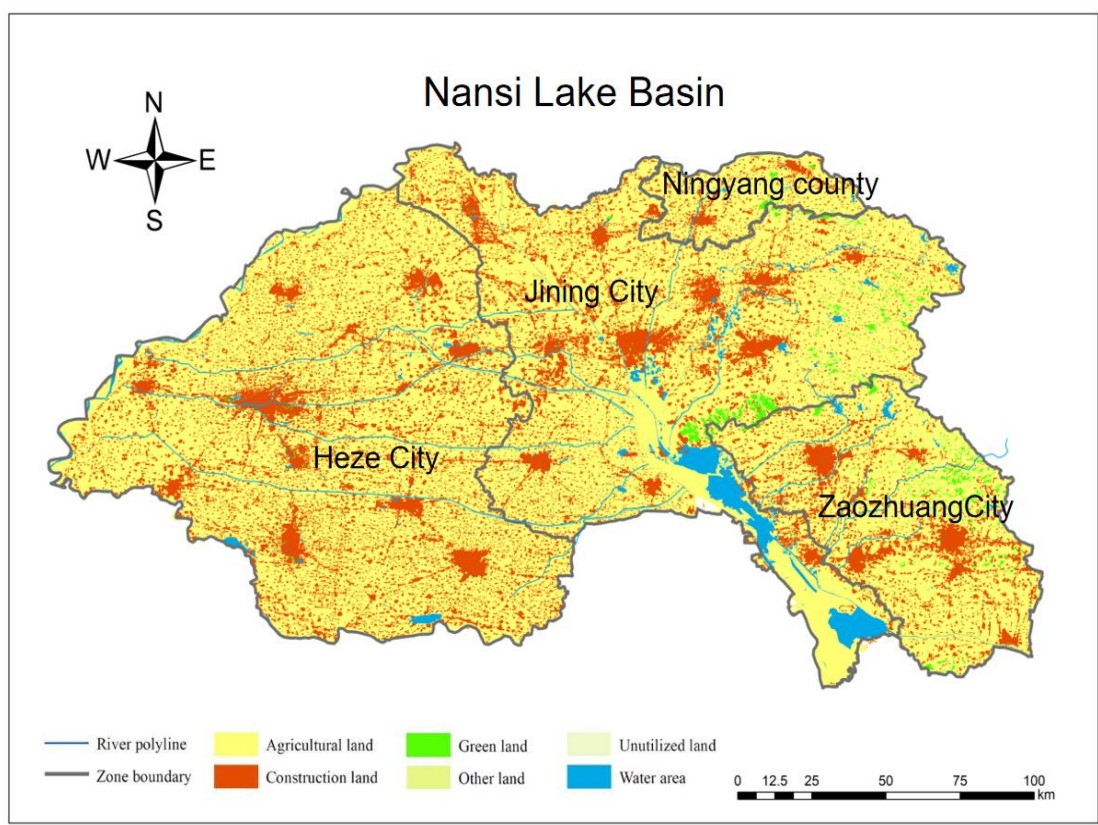

**Figure 1.** The study area.

Nansi Lake belongs to the Sishui River system in the Huaihe River, where 53 rivers from Shandong, Jiangsu, Henan and Anhui provinces converge. The lake is distributed in a zonal pattern in the northwest and southeast. Nansi Lake Basin has a temperate continental monsoon climate, with drought in spring and winter and rain in summer and autumn, and is greatly influenced by climate. In 2019, the annual average temperature is 14.5 °C and annual average precipitation is 856 mm. Due to its unique natural and geographical conditions, Nansi Lake Basin is rich in water resources, aquatic products and biological resources and is the largest freshwater lake, an important water supply place, a freshwater fishery base and a national ecological protection zone in Shandong Province. In addition, the Nansihu Lake Basin has the function of storing water sources and providing flow channels in China's South-to-North Water Transfer Project. The pillar industries in the basin include rice planting, breeding, paper making, food production and coal and electrical industry. From 2011 to 2019, during a period of rapid economic and social development, the resident population in the basin continued to grow, with 2.16 million people changing to 2.19 million. About 58% of these live in urban areas, and the gross domestic product (GDP) continues to increase, changing from 356.72 billion RMB to 641.60 billion RMB, of which the gross industrial output value accounts for 45% (Figure 2).

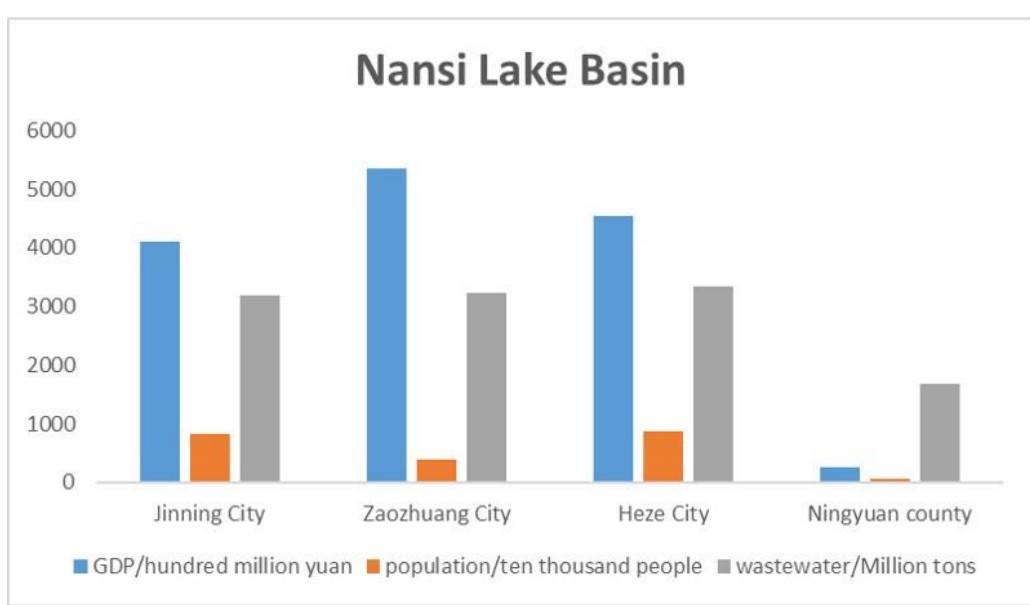

**Figure 2.** GDP, population and wastewater volumes in the Nansi Lake Basin.

However, there is an important impact on the ecological environment caused by high-intensity land development and inefficient land use brought on by rapid urbanization. The ecological land area of the watershed was reduced by 5% from 2011 to 2019, and the total waste water discharge was 994.109 million tons in 2019 and the total COD discharge was 140,728.27 tons. The occupation of these ecological lands and the discharge of pollutants pose a threat to the watershed ecosystem. For example, firstly, the Nansi Lake Basin is facing the challenge of shrinking water area, which will affect the water-holding capacity of the basin and the supply capacity of Shandong and even North China. Second, there is the risk of water quality fluctuation and water quality decline. According to data from 90 monitoring points in the Nansi Lake area from 2010 to 2017, the lake is seriously polluted and in a state of eutrophication. The main nutrient element is ammonia nitrogen, and the average ammonia nitrogen content in the estuary of the lake is 2.46 mg/L, which is far higher than the normal content (1 mg/L) and will pose risks to the safety of drinking water sources. In addition, the poor ability to resist climate disturbance leads to unstable water volume in Nansi Lake. In the past 20 years, the northern region has experienced continuous drought and little rain, and the total amount of water resources has declined. However, there is a contradiction between the ever-increasing water demand and the shortage of water resources in the basin. For example, agricultural water use is concentrated in spring and autumn, accounting for about 70% of the lake water.

In conclusion, in the context of economic development and population growth, the problems of occupation of ecological land and industrial pollutant emission in the watershed are becoming more and more serious, and the security of land use ecosystem is not certain. It is urgent to optimize land use allocation to balance the relationship between urban development and ecosystem protection, achieving the long-term sustainable development of the watershed. The optimal land use allocation model developed by this paper can deal with the uncertainty of the land use system, comprehensively consider the economic and ecological benefits of the land use ecosystem, protect ecological land from being occupied, control and reduce the emission of pollutants and finally realize sustainable development of the land use ecosystem. The model is applied to Nansi Lake Basin to obtain the optimal allocation scheme of sustainable land use, which provides reference for land managers to make regional development planning and demonstrates the feasibility and applicability of the model in planning sustainable land use under uncertain conditions.

*2.2. Data Sources*

Economic and social development data: population, gross domestic product, industrial output value and labor force are all from the 2009–2018 "Shandong Province Statistical Yearbook", "Jining City Statistical Yearbook", "Zaozhuang City Statistical Yearbook", "Heze City Statistical Yearbook" and "Ningyang County Statistical Yearbook".

Land use data: Remote sensing image data obtained from 30 m global surface cover data GlobeLand30 (http://www.globallandcover.com/, accessed on 18 January 2023). These were merged, cropped, interpreted and otherwise processed to obtain land use data.

DEM data: DEM data were downloaded from the geospatial data cloud platform of the Computer Network Information Center of the Chinese Academy of Sciences (http://www.gscloud.cn, accessed on 18 January 2023), and the slope and aspect data are processed.

Road and water system data: Shandong provincial roads, railways, highways, rivers, provincial boundaries and municipal boundaries and other data are sourced from the National Earth system science Data Center, and the road and water system data of the Nansihu Lake basin are cut out.

## 3. Interval Stochastic Fuzzy Chance-Constrained Programming Model

The established framework of the linear land use planning model with random fuzzy chance constraints in uncertain intervals (UISFC-LUP) is shown in Figure 3, and its establishment and solution process can be summarized as follows:

Firstly, a comprehensive analysis of the actual situation and characteristics of land use in the four basins of South Lake, combined with the necessary conditions required for development, was performed and economic and ecological objectives set as the objective functions of the model.

Secondly, the land use system is used as the basis for classifying the land in the study area into six types, namely agricultural land, construction land, grassland, watershed, other land and unused land. Focusing on the analysis based on the clarification of land use types, the analysis was combined with the characteristics of land use and economic development in the study area, and four major categories and 17 specific directions of economic, social, ecological and technological were selected as the constraints of the model with reference to the studies of other scholars.

Then, after considering the uncertainties in the land use system, coupling the interval parametric planning model, stochastic planning model and fuzzy elastic planning model, combining interval uncertainty, stochastic and fuzzy uncertainty with linear planning, a linear land use planning model with stochastic fuzzy chance constraints in uncertain intervals was established.

Finally, certain economic and environmental forecasting models and methods were used to obtain the model parameters through statistical yearbooks and EPA information; the parameters were substituted into the model and the model was solved through an interactive decomposition algorithm; finally, the results were analyzed to help land use planners and governments to make optimal decisions.

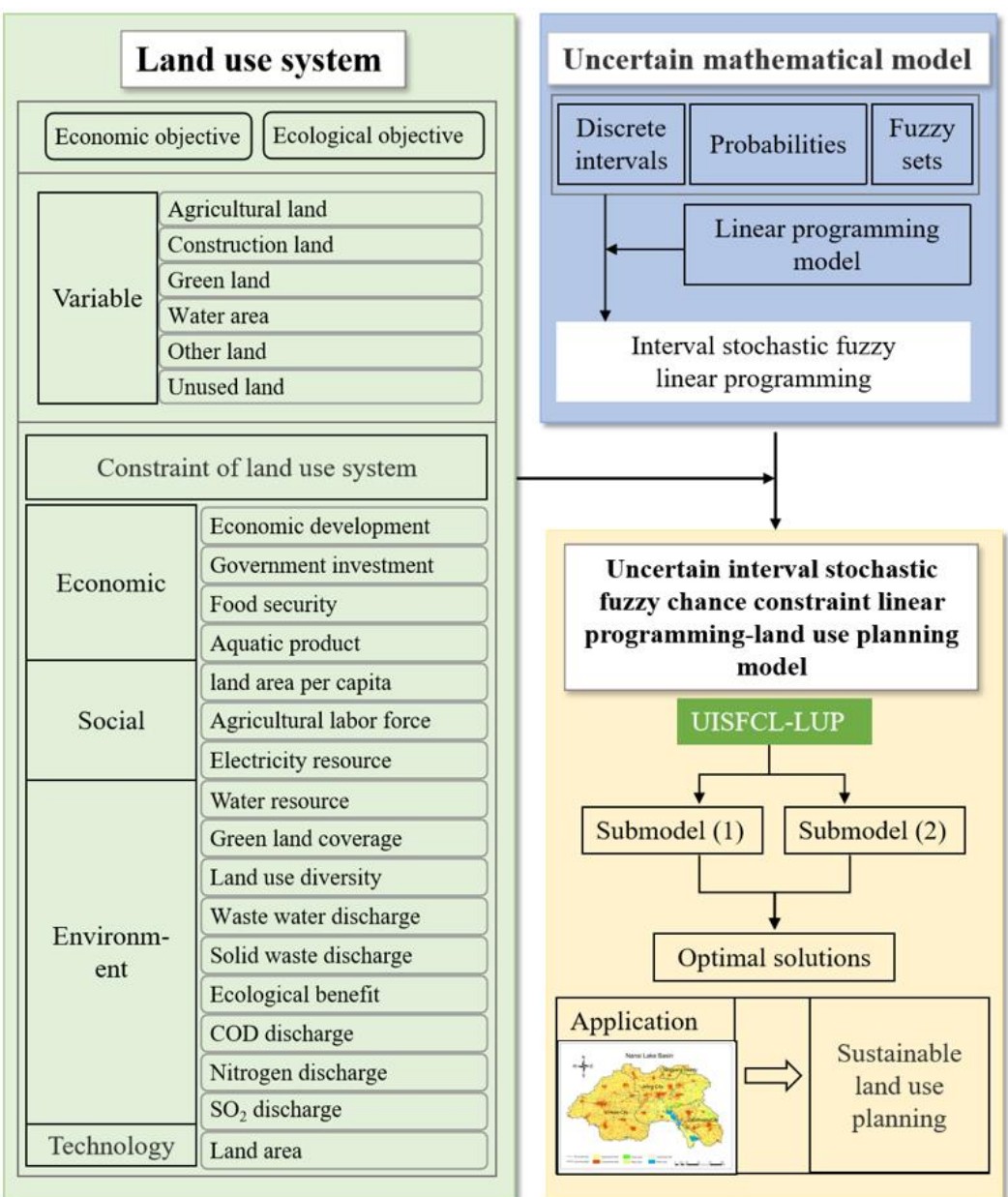

**Figure 3.** Framework of UISFCL-LUP.

The land use allocation model based on interval stochastic fuzzy chance constraints was established in this study. Under the condition of uncertain interval, the model can obtain the optimal allocation scheme of land use by establishing the objective function of land use system and a series of constraints. Six types of land use are included in the optimization process, which are agricultural land, construction land, green space, water area, other land and unused land. The purpose of the model is to achieve a sustainable program of land use systems with comprehensive consideration of economic development and ecological protection in the next two planning periods; the first is 2021–2025 and the second is 2025–2030. Many factors are considered in the model, which can be regarded as fitting into four categories: economic factors, social factors, environmental factors and technical factors. The specific model of this study is as follows:

(1)　Objective function: maximize the net system benefit

$$\textbf{\textit{Max}}\ \mathbf{Y}(\textbf{\textit{x}})^{\pm} \cong \sum_{i=1}^{4}\sum_{j=1}^{4}\sum_{t=1}^{2}\left(\mathbf{E}_{i,j,t}^{\pm}\times x_{i,j,t}^{\pm}\right) + \sum_{i=1}^{4}\sum_{j=1}^{4}\sum_{t=1}^{2}\left(\mathbf{ES}_{i,j,t}^{\pm}\times x_{i,j,t}^{\pm}\right) -$$

$$\sum_{i=1}^{4}\sum_{j=1}^{2}\sum_{t=1}^{2}\left(\mathbf{WUC}_{i,j,t}^{\pm}\times x_{i,j,t}^{\pm}\times \mathbf{WP} + \mathbf{EUC}_{i,j,t}^{\pm}\times x_{i,j,t}^{\pm}\times \mathbf{EP}\right) -$$

$$\sum_{i=1}^{4}\sum_{j=1}^{2}\sum_{t=1}^{2}\left(\mathbf{WTUC}_{i,j,t}^{\pm}\times x_{i,j,t}^{\pm}\right) - \sum_{i=1}^{4}\sum_{t=1}^{2}\left(\mathbf{STUC}_{i,j=2,t}^{\pm}\times x_{i,j=2,t}^{\pm}\right) -$$

$$\sum_{i=1}^{4}\sum_{t=1}^{2}\left(\mathbf{MUC}_{i,j=6,t}^{\pm}\times x_{i,j=6,t}^{\pm}\right) \tag{1}$$

where "$\pm$" means interval values; "$\cong$" means fuzzy equal; $Y(x_i)$ means net income of land use system during the planning period (RMB); x are variables which denotes the area of different land use (km$^2$); $i$ represents the different district, where $i = 1$ for Jining, $i = 2$ for Zaozhuang, $i = 3$ for Heze and $i = 4$ for Ningyang; $j$ means the type of land use, where $j = 1$ for agricultural land, $j = 2$ for construction land, $j = 3$ for green land, $j = 4$ for water land, $j = 5$ for other land, $j = 6$ for unused land; $t$ denotes time period of land use planning, where $t = 1$ for 2021–2025, $t = 2$ for 2025–2030.

The meaning of the objective function is as follows: during the planning period, the economic and ecological benefits generated by the agricultural land, construction land, green land and watershed in the four basins of South Lake minus the price of water, electricity, solid waste treatment, wastewater treatment and the maintenance cost required for unused land for agricultural and construction land, as shown in Tables 1–3.

**Table 1.** Constraint of ISFCL-LUP in this study.

| Index | Constrains | Formula | Description |
|---|---|---|---|
| (2) | Economic development constraints | $\dfrac{\sum_{i=1}^{4}\sum_{j=1}^{4}\left(E_{i,j,t+1}^{\pm}\times x_{i,j,t+1}^{\pm}\right)}{\sum_{i=1}^{4}\sum_{j=1}^{4}\left(E_{i,j,t}^{\pm}\times x_{i,j,t}^{\pm}\right)}\underset{\sim}{\geq}1$ | The economic benefit of the land use system during the planning period represents an increasing trend. |
| (3) | Government investment constraints | $\sum_{i=1}^{4}\sum_{j=1}^{2}\sum_{t=1}^{2}\left(\mathrm{WUC}_{i,j,t}^{\pm}\times x_{i,j,t}^{\pm}\times \mathrm{WP} + \mathrm{EUC}_{i,j,t}^{\pm}\times x_{i,j,t}^{\pm}\times \mathrm{EP}\right)$ $+\sum_{i=1}^{4}\sum_{j=1}^{2}\sum_{t=1}^{2}\left(\mathrm{WTUC}_{i,j,t}^{\pm}\times x_{i,j,t}^{\pm}\right)$ $+\sum_{i=1}^{4}\sum_{t=1}^{2}\left(\mathrm{STUC}_{i,j=2,t}^{\pm}\times x_{i,j=2,t}^{\pm}\right)$ $+\sum_{i=1}^{4}\sum_{t=1}^{2}\left(\mathrm{MUC}_{i,j=6,t}^{\pm}\times x_{i,j=6,t}^{\pm}\right)\underset{\sim}{\leq}\mathrm{MGI}_{i,t}^{\pm}$ | The operating cost of the land use system is funded by the government, but the cost shall not exceed it. |
| (4) | Food security constraints | $\sum_{i=1}^{4}\sum_{t=1}^{2}\mathrm{CUP}_{i,t}^{\pm}\times x_{i,j=1,t}^{\pm}\underset{\sim}{\geq}\sum_{i=1}^{4}\sum_{t=1}^{2}\mathrm{CD}_{i,t}^{\pm}$ | In order to ensure the realization of food security, the total grain output in the study area should meet the regional grain demand. |
| (5) | Aquatic product constraints | $\sum_{i=1}^{4}\sum_{t=1}^{2}\left(\mathrm{AUP}_{i,t}^{\pm}\times x_{i,j=4,t}^{\pm}\right)\underset{\sim}{\geq}\sum_{i=1}^{4}\sum_{t=1}^{2}\left(\mathrm{AD}_{i,t}^{\pm}\times \mathrm{TP}_{i,t}^{\pm}\right)$ | In order to answer people's need for aquatic products, the production of aquatic products in the study area should be more than the need for aquatic products. |
| (6) | Land area per capita constraints | $\dfrac{\sum_{i=1}^{4}\sum_{t=1}^{2}\mathrm{TP}_{i,t}^{\pm}}{\sum_{i=1}^{4}\sum_{j=1}^{6}\sum_{t=1}^{2}x_{i,j,t}^{\pm}}\underset{\sim}{\leq}\sum_{i=1}^{4}\sum_{t=1}^{2}\mathrm{PCL}_{i,t}^{\pm}$ | Realizing the rational use of land, the model should ensure that the per-person land occupation area is greater than or equal to the minimum per-person land occupation area. |
| (7) | Water resource constraints | $\sum_{i=1}^{4}\sum_{t=1}^{2}\left(\mathrm{WUC}_{i,j=1,t}^{\pm}\times x_{i,j=1,t}^{\pm}\right)$ $+\sum_{i=1}^{4}\sum_{t=1}^{2}\left(\mathrm{WUC}_{i,j=2,t}^{\pm}\times x_{i,j=2,t}^{\pm}\right)$ $\underset{\sim}{\leq}\sum_{i=1}^{4}\sum_{t=1}^{2}\mathrm{AW}_{i,t}^{\pm}$ | Water resources are needed by the operation of land use system, while due to the limitation of water supply in the basin, the total water usage of the land use system should not exceed the usable water usage |
| (8) | Electricity resource constraints | $\sum_{i=1}^{4}\sum_{t=1}^{2}\left(\mathrm{EUC}_{i,j=1,t}^{\pm}\times x_{i,j=1,t}^{\pm}\right)$ $+\sum_{i=1}^{4}\sum_{t=1}^{2}\left(\mathrm{EUC}_{i,j=2,t}^{\pm}\times x_{i,j=2,t}^{\pm}\right)\underset{\sim}{\leq}\sum_{i=1}^{4}\sum_{t=1}^{2}\mathrm{EW}_{i,t}^{\pm}$ | The operation of the land use system requires electricity. Due to the limitation of power supply in the river basin, the total electricity usage of the land use system shall not exceed the usable electricity. |

**Table 1.** *Cont.*

| Index | Constrains | Formula | Description |
|---|---|---|---|
| (9) | Agricultural labor force constraints | $\sum\limits_{i=1}^{4}\sum\limits_{t=1}^{2}\left(\text{LUA}_{i,t}^{\pm}\times x_{i,j=1,t}^{\pm}\right)\underset{\sim}{\leq}\sum\limits_{i=1}^{4}\sum\limits_{t=1}^{2}\text{ALA}_{i,t}^{\pm}$ | In order to cope with the trend of non-agriculturalization of the agricultural population, the amount of labor required for agricultural production should not be higher than that of agricultural labor. |
| (10) | Agricultural land area constraints | $\sum\limits_{i=1}^{4}\sum\limits_{t=1}^{2}x_{i,j=1,t}^{\pm}\underset{\sim}{\geq}\sum\limits_{i=1}^{4}\sum\limits_{t=1}^{2}\text{AM}_{i,t}^{\pm}$ | Based on the national requirements of "cherishing and rationally utilizing land and earnestly protecting cultivated land", the area of cultivated land shall not be inferior to the planning requirements. |
| (11) | Green land coverage constraints | $\dfrac{\sum_{i=1}^{4}\sum_{t=1}^{2}x_{i,j=3,t}^{\pm}}{\sum_{i=1}^{4}\sum_{j=1}^{6}\sum_{t=1}^{2}x_{i,j,t}^{\pm}}\underset{\sim}{\geq}\sum\limits_{i=1}^{4}\sum\limits_{t=1}^{2}\text{FR}_{i,t}^{\pm}$ | The protection of green space is still challenged, so the coverage rate of green space in river basins should exceed or be equal to the planning requirements. |
| (12) | Land use diversity constraints | $-\dfrac{\sum_{j=3}^{4}x_{i,j,t}^{\pm}}{\text{TA}_{i,t}^{\pm}}\times In\left(\dfrac{\sum_{j=3}^{4}x_{i,j,t}^{\pm}}{\text{TA}_{i,t}^{\pm}}\right)-\dfrac{x_{i,j=2,t}^{\pm}}{\text{TA}_{i,t}^{\pm}\times}In\left(\dfrac{x_{i,j=2,t}^{\pm}}{\text{TA}_{i,t}^{\pm}}\right)\underset{\sim}{\geq}\text{SHDI}_{i,t}^{\pm}$ | Diversity should be one of the characteristics of the landscape of land use system, so the land use in the model should meet the requirements of diversity. |
| (13) | Waste water discharge constraints | $\sum\limits_{i=1}^{3}\sum\limits_{j=1}^{2}\sum\limits_{t=1}^{2}\left(\text{WUD}_{i,j,t}^{\pm}\times x_{i,j,t}^{\pm}\right)\underset{\sim}{\leq}\sum\limits_{i=1}^{4}\sum\limits_{t=1}^{2}\text{AWC}_{i,t}^{\pm}$ | The wastewater discharge amount of the land use system should not be higher than the wastewater treatment capacity in river basins. |
| (14) | Solid waste discharge constraints | $\sum\limits_{i=1}^{3}\sum\limits_{j=1}^{2}\sum\limits_{t=1}^{2}\left(\text{SUD}_{i,j,t}^{\pm}\times x_{i,j,t}^{\pm}\right)\underset{\sim}{\leq}\sum\limits_{i=1}^{4}\sum\limits_{t=1}^{2}\text{ASC}_{i,t}^{\pm}$ | The amount of solid waste discharged from the land use system shall not be higher than the capacity of solid waste treatment in the basin. |
| (15) | Ecological benefit constraints | $\dfrac{\sum_{i=1}^{4}\sum_{j=1}^{6}\left(\text{ES}_{i,j,t+1}^{\pm}\times x_{i,j,t+1}^{\pm}\right)}{\sum_{i=1}^{4}\sum_{j=1}^{6}\left(\text{ES}_{i,j,t}^{\pm}\times x_{i,j,t}^{\pm}\right)}\underset{\sim}{\geq}1$ | The value of the watershed ecological service is an important part of the regional economy, and realizing the growth of ecological benefits is an important means to ensure good ecology. The increasing trend of ecological benefits can be written as follows. |
| (16) | COD discharge constraints | $\sum\limits_{i=1}^{4}\sum\limits_{t=1}^{2}\left(\text{COD}_{i,j=2,t}^{\pm}\times x_{i,j=2,t}^{\pm}\right)\underset{\sim}{\leq}\sum\limits_{i=1}^{4}\sum\limits_{t=1}^{2}\text{DC}_{i,t}^{\pm}$ | Water resources are the core of the basin environment, which is why it is vital to control the emission of COD to ensure water quality safety. Therefore, COD emissions should meet the planning requirements. |
| (17) | Nitrogen discharge constraints | $\sum\limits_{i=1}^{4}\sum\limits_{t=1}^{2}\left(N_{i,j=2,t}^{\pm}\times x_{i,j=2,t}^{\pm}\right)\underset{\sim}{\leq}\sum\limits_{i=1}^{4}\sum\limits_{t=1}^{2}\text{DN}_{i,t}^{\pm}$ | There is an important correlation between nitrogen and water quality. For the improvement of the eutrophication status of Nansi Lake, it is vital to control the emission of nitrogen. Therefore, the total amount of ammonia nitrogen emissions from the land use system should be less than the planned control. |
| (18) | SO₂ discharge constraints | $\sum\limits_{i=1}^{4}\sum\limits_{t=1}^{2}\left(\text{SO2}_{i,j=2,t}^{\pm}\times x_{i,j=2,t}^{\pm}\right)\underset{\sim}{\leq}\sum\limits_{i=1}^{4}\sum\limits_{t=1}^{2}\text{DS}_{i,t}^{\pm}$ | Sulfur dioxide is the main air pollutant produced by construction land, so it is necessary to control the emission of sulfur dioxide. According to regional environmental planning, sulfur dioxide discharge amount should be lessened yearly. |
| (19) | Land area constraints | $\sum\limits_{i=1}^{3}\sum\limits_{j=1}^{6}\sum\limits_{t=1}^{2}x_{i,j,t}^{\pm}=\sum\limits_{i=1}^{4}\sum\limits_{t=1}^{2}\text{TA}_{t}^{\pm}$ | The optimized allocation area of the model shall not exceed the planned area. |
| (20) | Non-negative constraints | $x_{i,j,t}^{\pm}>0$ | The area of land use type cannot be negative. |

**Table 2.** The unit economic benefit of land use.

| Name (10⁶ RMB/km²) | T = 1 | | T = 2 | |
|---|---|---|---|---|
| | Lower | Upper | Lower | Upper |
| $E_{i=1,j=1}^{\pm}$ | 41.85 | 56.62 | 46.93 | 63.49 |
| $E_{i=1,j=2}^{\pm}$ | 1702.74 | 2303.71 | 2045.05 | 2766.83 |
| $E_{i=1,j=3}^{\pm}$ | 86.45 | 116.96 | 91.96 | 124.42 |
| $E_{i=1,j=4}^{\pm}$ | 13.64 | 18.46 | 16.04 | 21.70 |

**Table 3.** The unit ecological benefit of land use.

| NAME (10$^6$ RMB/km$^2$) | Lower | Upper |
|---|---|---|
| $ES^{\pm}_{j=1}$ | 2.6 | 3.5 |
| $ES^{\pm}_{j=2}$ | 0.2 | 0.3 |
| $ES^{\pm}_{j=3}$ | 17.45 | 23.65 |
| $ES^{\pm}_{j=4}$ | 8.1 | 11 |

There are many existing land use problems in the four watersheds of South Lake, such as a large number of people and a small amount of land, insufficient land supply, serious deterioration of the land environment, low levels of economical and intensive use, insufficient arable land and serious ecological damage. Based on the above realistic conditions and combined with previous scholars' research, 17 constraints were selected after considering the influencing factors of the existing land use problems, as shown in Table 4.

**Table 4.** Socio-economic and environmental parameters.

| NAME (RMB/km$^2$) | Unit | T = 1 | | T = 2 | |
|---|---|---|---|---|---|
| | | Lower | Upper | Lower | Upper |
| $MUC^{\pm}$ | 10$^4$ RMB/km$^2$ | 637 | 812.5 | 702 | 955.5 |
| $WTUC^{\pm}_{j=1}$ | 10$^3$ RMB/km$^2$ | 115.32 | 126.31 | 130.3116 | 142.7303 |
| $WTUC^{\pm}_{i=1,j=2}$ | 10$^3$ RMB/km$^2$ | 498.03 | 530.595 | 568.49 | 586.56 |
| $WTUC^{\pm}_{i=2,j=2}$ | 10$^3$ RMB/km$^2$ | 365.69 | 401.90 | 465.60 | 488.93 |
| $WTUC^{\pm}_{i=3,j=2}$ | 10$^3$ RMB/km$^2$ | 474.24 | 509.80 | 539.89 | 612.30 |
| $WTUC^{\pm}_{i=4,j=2}$ | 10$^3$ RMB/km$^2$ | 42.67 | 45.87 | 48.58 | 55.09 |
| $STUC^{\pm}_{i=1}$ | 10$^3$ RMB/km$^2$ | 4.75 | 5.14 | 5.59 | 6.11 |
| $STUC^{\pm}_{i=2}$ | 10$^3$ RMB/km$^2$ | 3.71 | 3.90 | 4.68 | 5.20 |
| $STUC^{\pm}_{i=3}$ | 10$^3$ RMB/km$^2$ | 4.36 | 4.55 | 5.46 | 5.85 |
| $STUC^{\pm}_{i=4}$ | 10$^3$ RMB/km$^2$ | 0.39 | 0.41 | 0.49 | 0.53 |
| $WUC^{\pm}_{i=1,j=1}$ | 10$^4$ m$^3$/km$^2$ | 95.67 | 129.44 | 77.76 | 105.20 |
| $EUC^{\pm}_{i=1,j=1}$ | 10$^4$ kWh/km$^2$ | 139.45 | 188.67 | 161.45 | 218.43 |
| $WP^{\pm}_{j=1}$ | RMB/m$^3$ | 0.61 | | 0.83 | |
| $CUP^{\pm}_{i=1}$ | 10$^2$ ton/km$^2$ | 43.55 | 58.92 | 48.84 | 66.08 |
| $CD^{\pm}$ | 10$^2$ ton/people | 81.75 | 110.60 | 66.23 | 89.61 |
| $AUP^{\pm}_{i=1}$ | 10$^2$ ton/km$^2$ | 25.36 | 34.30 | 26.79 | 36.25 |
| $AD^{\pm}$ | 10$^2$ ton/people | 11.23 | 15.19 | 12.60 | 17.05 |
| $TP^{\pm}_{i=1}$ | 10$^4$ people | 3674.57 | 4971.48 | 3767.71 | 5097.49 |
| $PCL^{\pm}_{t}$ | People/km$^2$ | 200 | | 400 | |
| $AW^{\pm}_{i=1}$ | 10$^8$ m$^3$ | 83.91 | 113.52 | 74.28 | 100.50 |
| $LUA^{\pm}$ | people | 153.00 | | 207.00 | |
| $ALA^{\pm}_{i=1}$ | 10$^4$ people | 1538.11 | 2080.97 | 762.90 | 1032.16 |
| $FR^{\pm}_{i}$ | % | 22.95 | | 31.05 | |
| $SHDI^{\pm}_{i=1}$ | | 0.44 | | 0.60 | |
| $WUD^{\pm}_{i=1,j=1}$ | ton/km$^2$ | 139.12 | 188.23 | 167.28 | 226.33 |
| $WUD^{\pm}_{i=1,j=2}$ | 10$^4$ ton/km$^2$ | 27.86 | 37.69 | 12.90 | 17.45 |
| $SUD^{\pm}_{i=1,j=2}$ | ton/km$^2$ | 30,322.77 | 41,024.93 | 10,880.90 | 14,721.22 |
| $ASC^{\pm}_{i=1}$ | 10$^4$ ton | 3929.10 | 5315.84 | 1126.58 | 1524.20 |
| $COD^{\pm}_{i=1,j=2}$ | ton/km$^2$ | 128.27 | 173.54 | 130.56 | 176.64 |
| $DC^{\pm}_{i=1}$ | ton | 124,341.94 | 168,227.33 | 66,745.37 | 90,302.56 |
| $N^{\pm}_{i=1,j=1}$ | ton/km$^2$ | 2.36 | 3.20 | 1.31 | 1.78 |
| $N^{\pm}_{i=1,j=2}$ | ton/km$^2$ | 23.01 | 31.13 | 24.75 | 33.49 |
| $DN^{\pm}_{i=1}$ | ton | 19,037.24 | 25,756.27 | 9272.43 | 12,545.05 |
| $SO2^{\pm}_{i=1,j=2}$ | ton/km$^2$ | 69.99 | 94.69 | 25.65 | 34.71 |
| $DS^{\pm}_{i=1}$ | ton | 122,540.01 | 165,789.42 | 59,685.30 | 80,750.70 |

## 4. Result Analysis

Each development area corresponds to different results of optimal allocation of land use. Based on the model calculation, we obtained the optimal allocation results of six land use types in four regions based on different environmental constraints.

### 4.1. Scheme of Land Use Optimal Allocation Model

As expressed in Figure 4, through model calculation, we get the optimal allocation scheme of land use under different $p$ values. Different $p$ levels represent the severity of environmental constraints. The smaller $p$ value represents more stringent environmental conditions, which can ensure the requirements of ecological balance and environmental protection and produce more conservative land use policies.

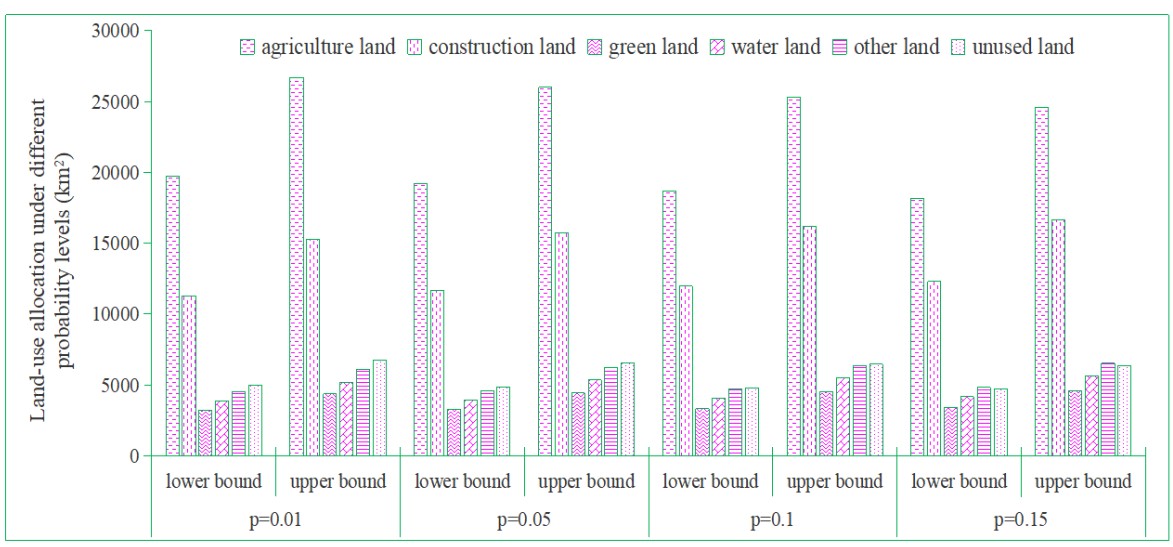

**Figure 4.** Land use allocation under different $p$ levels.

For example, under the condition of $p = 0.01$, the probability of violating environmental constraints is the smallest and the areas of agricultural land, construction land, green land, water area, other land and unused land in the optimal allocation of land use in restricted development areas are [19,735.88, 26,701.49] km$^2$, [11,280.38, 15,261.69] km$^2$, [3217.09, 4352.53] km$^2$, [3841.76, 5197.67] km$^2$, [4488.94, 6073.28] km$^2$, [4980.61, 6738.47] km$^2$ and the land use benefit is [18.4, 20.4] × 1012 RMB, which means that the system benefit will vary during [18.4, 20.4] × $10^{12}$ RMB. This means that the actual value of each continuous variable changes within its lower and upper limits. Specifically, the lower-limit return (f(x) = 18.4 × $10^{12}$ RMB) matches the lower-limit of the decision variable value ($x_j = 1 = 19{,}735.88$, $x_j = 2 = 11{,}280.38$, $x_j = 3 = 3217.09$, $x_j = 4 = 3841.76$, $x_j = 5 = 4488.94$, $x_j = 6 = 4980.61$, km$^2$). In contrast, the upper limit system benefit (f(x) = 20.4 × $10^{12}$ RMB) will be equal to the upper limit of decision variable value ($x_j = 1 = 26{,}701.49$, $x_j = 2 = 15{,}261.69$, $x_j = 3 = 4352.53$, $x_j = 4 = 5197.67$, $x_j = 5 = 6073.28$, $x_j = 6 = 6738.47$, km$^2$). Generally, there is a low risk of violating system constraints in land use strategies with low returns. Conversely, the utilization strategy aiming at higher income will be more likely to violate the system constraints.

When the land use mode is selected according to the land use area value in the interval between the upper limit and the lower limit, the system benefit value will change synchronously at the upper and lower limits. Thus, according to the actual situation in the planning period, the land use allocation scheme can be obtained by changing different land use combinations. The change of system conditions caused by the existence of realistic uncertainty can be expressed powerfully. The elastic interval of decision variables provided by the UISFCL-LUP solution can help decision makers directly improve the scientificity of alternatives or adjust the decision variables to get more supportive decision plans.

Therefore, UISFCL-LUP method allows decision makers to integrate tacit knowledge into the model, so as to obtain scientific and applicable decision-making results.

### 4.2. Relationship between Land Use Structure and Regional Development Strategy

Figure 5 shows the optimal allocation of agricultural land in four districts over two planning periods. In the first planning period, the optimal allocation area of agricultural land in the restricted development area is [19,735.88, 26,701.49] km$^2$, in the core protection area [8502.41, 11,503.27] km$^2$, in the controlled development area [29,397.06, 39,772.49] km$^2$ and in the core development area [2438.44, 3299.06] km$^2$. In the second planning period, the optimal allocation area of restricted development areas is [19,341.17, 26,167.46] km$^2$, in the core protection area [8247.34, 11,158.17] km$^2$, in the restricted development area [28,221.17, 38,181.59] km$^2$ and in the key protected area [2316.52, 3134.11] km$^2$.

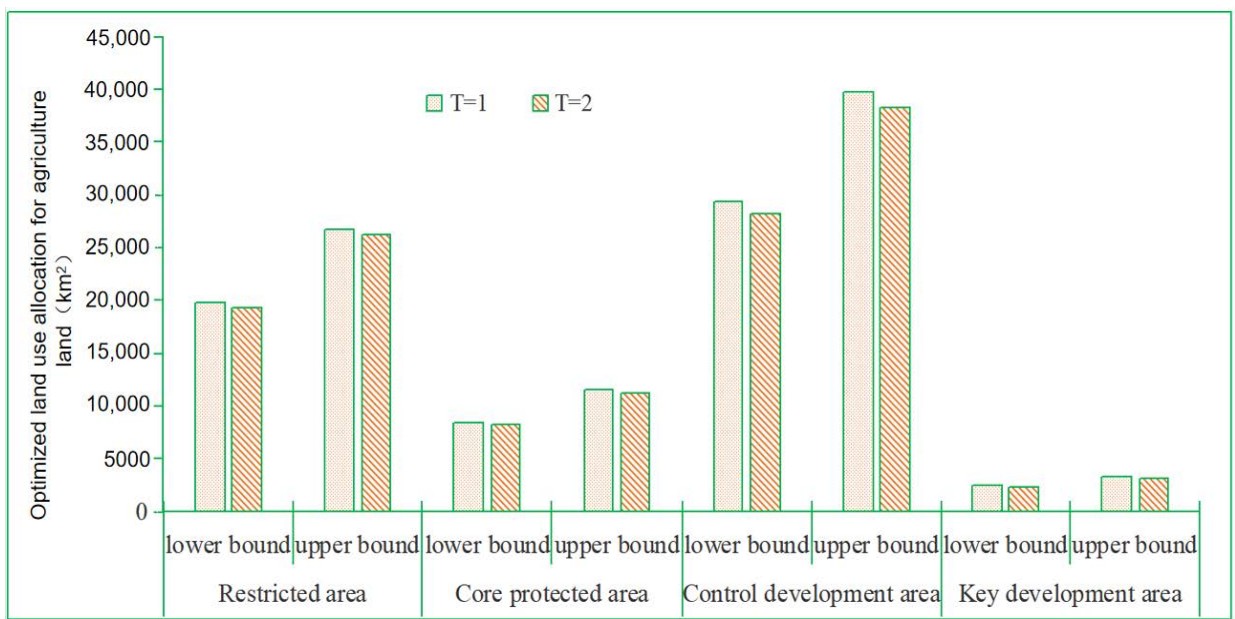

**Figure 5.** Optimized land use allocation for agriculture land.

Figure 6 shows the optimal allocation of construction land in four districts over two planning periods. In the first planning period, the optimal allocation area of construction land in the restricted development area is [11,280.38, 15,261.69] km$^2$, in the core protection area [4605.35, 6230.76] km$^2$, in controlled development area [12,076.22, 16,338.41] km$^2$ and in the core development area [958.55, 1296.86] km$^2$. In the second planning period, the optimal allocation area of the restricted development area is [11,557.20, 15,636.22] km$^2$, that of core protection area is [4818.42, 6519.04] km$^2$, that of the restricted development area is [28,221.17, 38,181.59] km$^2$ and that of the key development area is [1037.80, 1404.08] km$^2$.

Figure 7 shows the optimal allocation of green space in four districts over two planning periods. In the first planning period, the optimal allocation area of green space in restricted development area is [3217.09, 4352.53] km$^2$, in the core protection is [632.38, 855.57] km$^2$, in the controlled development area [2779.37, 3760.32] km$^2$ and in the core development area [285.54, 386.31] km$^2$. In the second planning period, the optimal allocation area of the restricted development area is [3261.58, 4412.72] km$^2$, that of the core protection area is [658.77, 891.28] km$^2$, that of the restricted development area is [2888.99, 3908.63] km$^2$ and that of the key protection area is [298.27, 403.55] km$^2$.

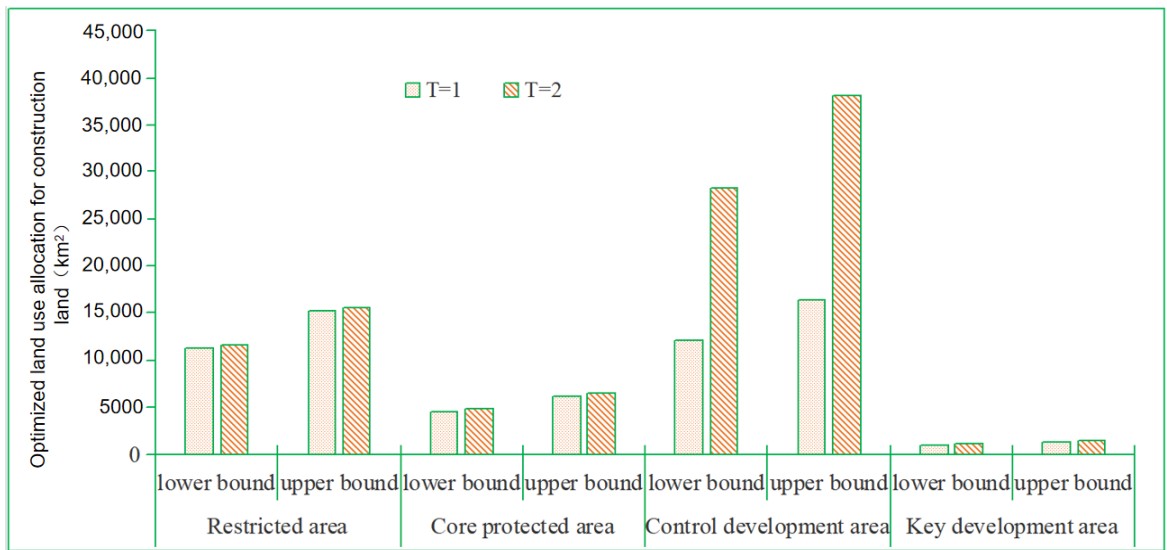

**Figure 6.** Optimized land use allocation for construction land.

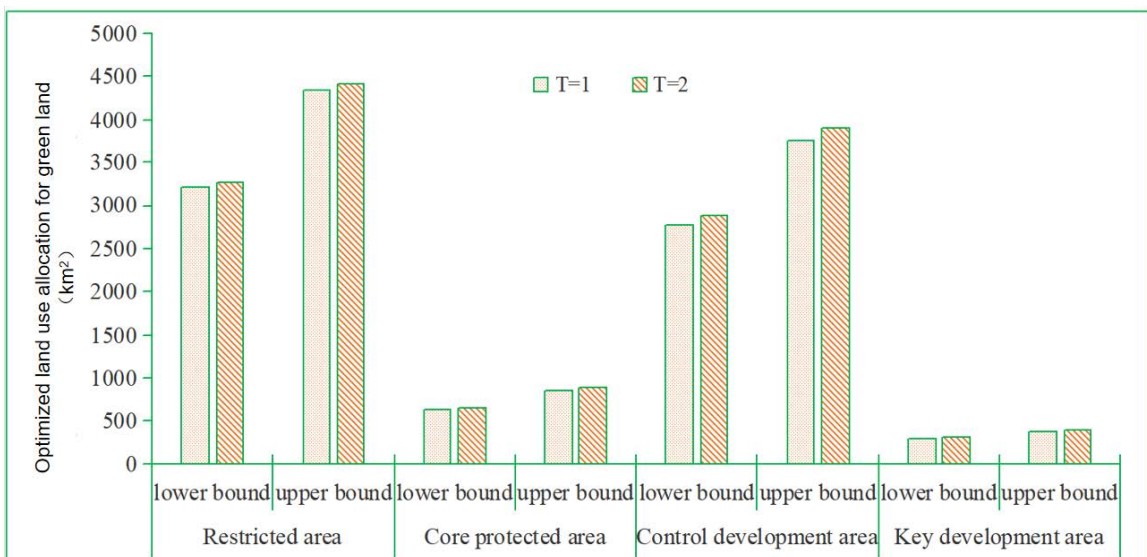

**Figure 7.** Optimized land use allocation for green land.

Figure 8 shows the optimal allocation of water areas in four districts over two planning periods. In the first planning period, the optimal allocation area of water in the restricted development area is [3841.76, 5197.67] km$^2$, in the core protection area [828.59, 1121.03] km$^2$, in the controlled development area [2779.37, 3760.32] km$^2$ and in the core development area [279.03, 377.52] km$^2$. In the second planning period, the optimal allocation area of restricted development areas is [3925.79, 5311.37] km$^2$, that of the core protection area is [878.43, 1188.47] km$^2$, that of the restricted development area is [1988.72, 2690.62] km$^2$ and that of the key development area is [303.09, 410.07] km$^2$.

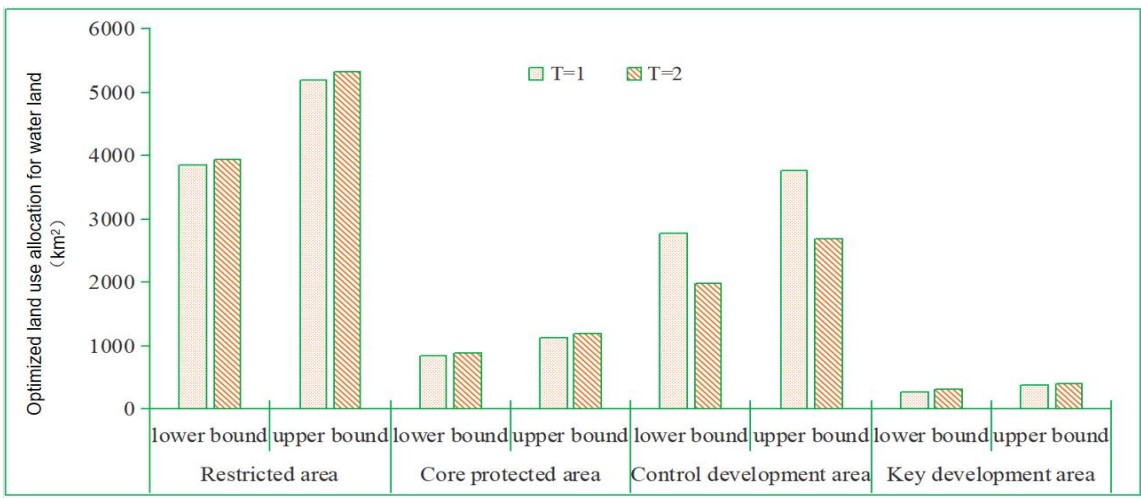

**Figure 8.** Optimized land use allocation for water land.

Figure 8 shows that the optimal allocation of water areas in four districts over two planning periods. In the first planning period, the optimal allocation area of water in the restricted development area is [3841.76, 5197.67] km², in the core protection area [828.59, 1121.03] km², in the controlled development area [2779.37, 3760.32] km² and in the core development area [279.03, 377.52] km². In the second planning period, the optimal allocation area of the restricted development area is [3925.79, 5311.37] km², that of the core protection area is [878.43, 1188.47] km², that of the restricted development area is [1988.72, 2690.62] km² and that of the key development area is [303.09, 410.07] km².

Figures 9 and 10 show the optimal allocation of unused land and other land in four districts over two planning periods. In the first planning period, the optimal allocation areas of other land and unused land in the restricted development areas are [4488.94, 6073.28] km² and [4980.61, 6738.47] km², while the optimal allocation areas in the core protection area are [3555.11, 4809.85] km² and [1271.17, 1271.17] km². The areas in the controlled development areas are [4572.45, 6186.25] km² and [1052.98, 1424.62] km², while the areas in the core development areas are [427.64, 578.57] km² and [392.06, 530.44] km². In the second planning period, the optimal allocation areas of the restricted development areas are [4577.92, 6193.66] km² and [4881.00, 6603.70] km², while the optimal allocation areas of the core protection area are [3607.88, 4881.25] km² and [1233.03, 1668.22] km², in controlled development area 6482.87 km² and [1010.86, 1367.64] km² and the areas in key development areas [453.11, 613.03] km² and [372.46, 503.92] km².

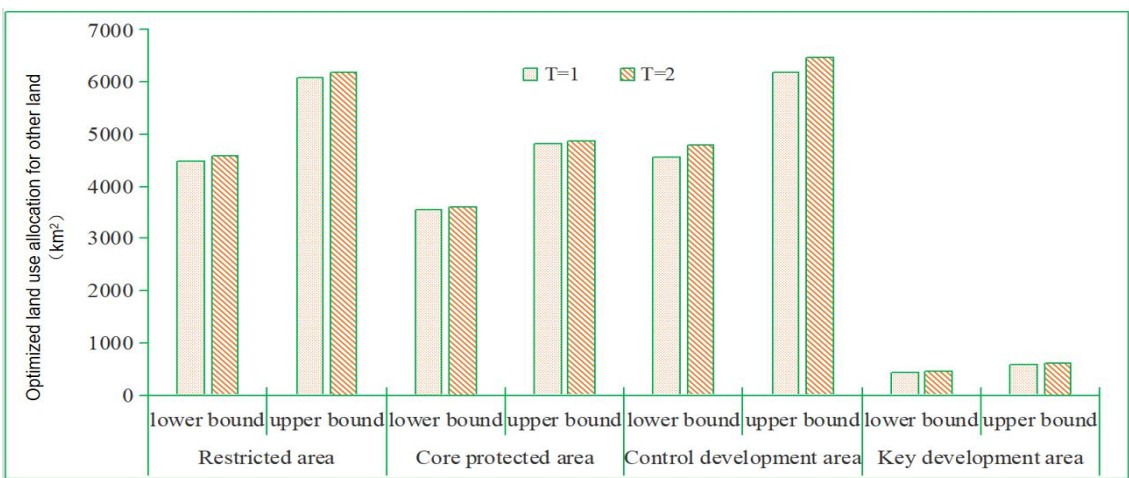

**Figure 9.** Optimized land use allocation for other land.

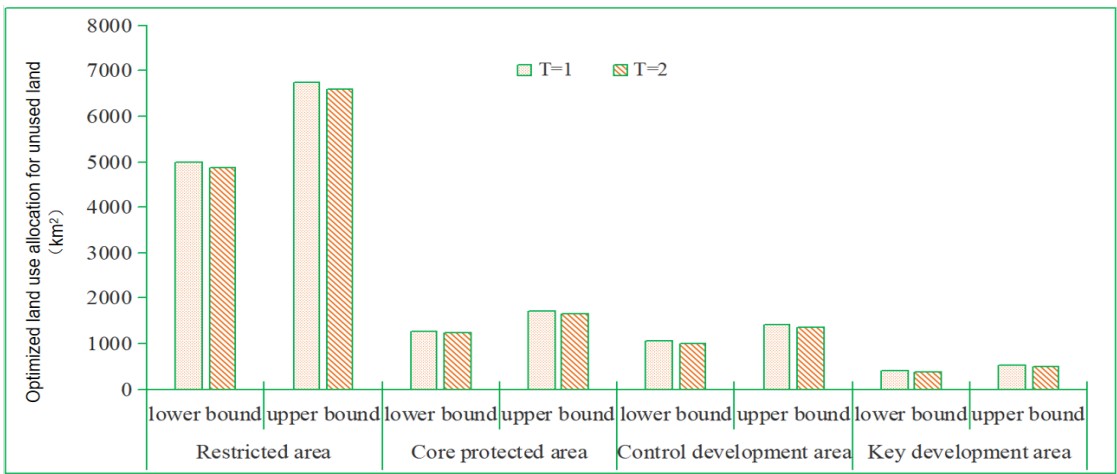

**Figure 10.** Optimized land use allocation for unused land.

### 4.3. Relationship between System Constraints and System Benefits

In the model, the value of λ indicates the probability of meeting the optimization objectives and constraints under specific system conditions. As shown in Figure 11, according to the model results, the value of λ is [0.54, 0.79]. When λ tends to the lower bound, it is more consistent with the constraints and conditions of the land use system, which implies a more conservative land use strategy. When λ tends to the upper bound, it indicates that the land use strategy is more active. It can be seen from the figure that there is a strong positive correlation between λ and system benefits. The optimal allocation model is based on uncertain conditions and realizes the optimization objectives and constraints of land use system. When the model reaches the maximum satisfaction of the land use system, λ is 0.54 and the corresponding benefit of land use system is [18.4, 20.4] × 10$^{12}$ RMB. When λ is 0.79, the corresponding benefit of land use system is [96.7, 109.3] × 10$^{12}$ RMB.

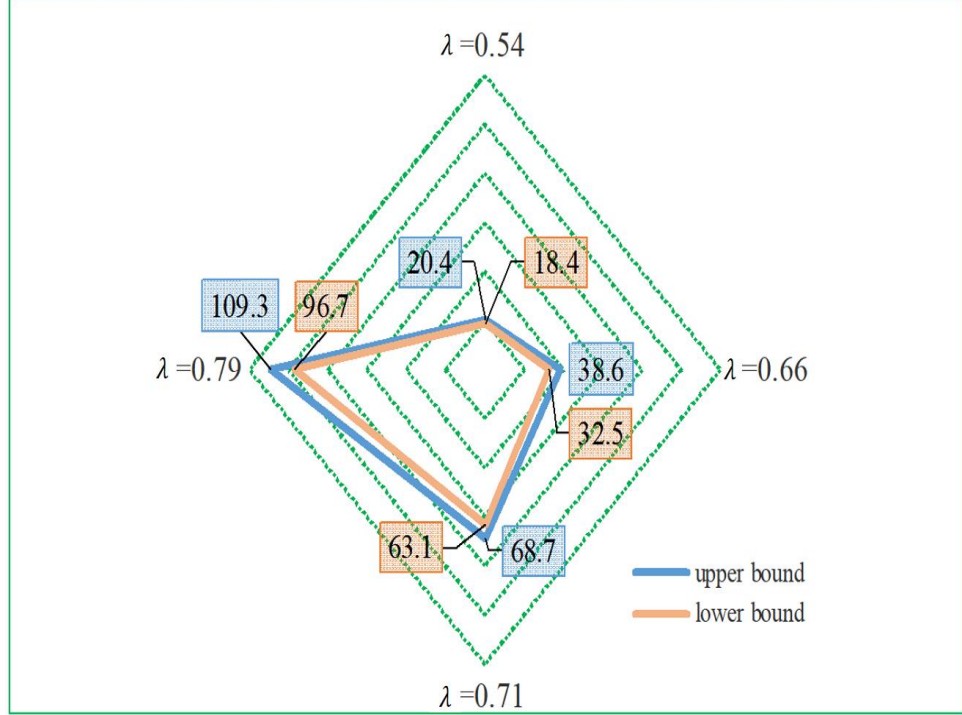

**Figure 11.** System benefits under different λ levels.

Land managers can make different decisions based on realistic conditions and the interval number of λ. The value of λ indicates that the interests and constraints of the

land use system compete with each other. The smaller value of λ means that the model will better meet the needs of social and environmental development and protection and corresponds to a more conservative land use strategy. Similarly, a larger value of λ means that the model will favor more system benefits, which correspond to a more ambitious land use strategy.

### 4.4. Relationship between Environmental Restriction and System Benefit

In the model, $p$ value represents the degree of environmental constraints. A smaller $p$ value represents more stringent environmental conditions, which can ensure that the requirements of ecological balance and environmental protection are met and produce more conservative land use policies. Similarly, the greater the $p$ value, the greater the possibility of violating environmental constraints and the more optimistic the land use policy.

As shown in Figures 4 and 12, when $p$ = 0.01, the optimal allocation area of land use is [19,735.88, 26,701.49] km$^2$, [11,280.38, 15,261.69] km$^2$, [3217.09, 4352.53] km$^2$ and [3841.76, 5197.67] km$^2$, respectively, and the system benefit is [18.4, 20.4] $\times 10^{12}$ RMB. When $p$ = 0.05, the optimal allocation area of land use changes to [19,216.52, 25,998.82] km$^2$, [11,646.61, 15,757.17] km$^2$, [3275.95, 4432.16] km$^2$ and [3952.93, 5348.09] km$^2$respectively, and the system benefit is [32.5, 38.6] $\times 10^{12}$ RMB. The figure shows the relationship between $p$ value and system benefits. The balance between land use system benefits, that is, functional objectives and environmental constraints, can be reflected by the value of $p$. The figure shows that there is a positive correlation between system benefits and $p$ value. In addition, as the $p$ value increases, there is a greater possibility of violating environmental constraints; at the same time, the stringency of the constraints is reduced and the system benefits increase, corresponding to more optimistic land use decisions.

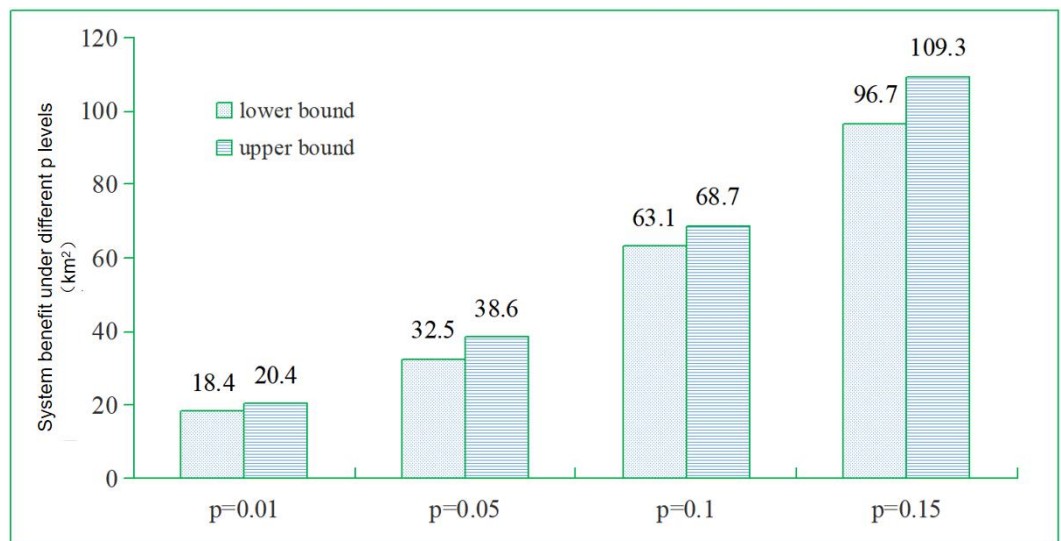

**Figure 12.** System benefit under different $p$ levels.

## 5. Conclusions and Discussion

### 5.1. Main Conclusions

By combining the uncertain mathematical model with land use planning, an optimal method of land use allocation with uncertain interval random fuzzy chance constraints is constructed. UISFCL-LUP can deal with the uncertainty of interval, membership function and probability representation and can also be used to solve land use planning and land use strategy analysis under uncertain conditions. In addition, the model also considers the constraints of social factors, environmental factors and ecological conditions and can quantitatively express the inverse relationship between economic development and environmental protection in the land use system [64].

This method has excellent performance in the optimal allocation of land use in Nansi Lake Basin and has certain practical applicability. According to the results of the model, the model gives a series of interval values under different environmental conditions. Land managers can make judgments according to different social and economic development needs in different regions and determine the strategic land use allocation scheme under uncertain conditions. At the same time, the model obtains interval solutions under different system satisfaction and probability of violating constraints, which is helpful for land managers to have a deeper analysis of the importance of optimal development and sustainable development of land system [65].

*5.2. Disscusion*

The UISFCL-LUP method is an aggregation of interval parametric programming, fuzzy linear programming and chance constrained programming, which can handle uncertain problems such as interval value, fuzzy set and probability and has the following advantages. First of all, it can scientifically predict the future state of land use system and socio-economic situation based on forecasting method. Secondly, it takes the economic development benefit, ecological benefit and land use cost of land use system into the objective function, which maintains the common development of economy and ecological environment. In addition, it uses the interval stochastic fuzzy programming model to solve the problem of quantitative land area optimization, which can obtain a sustainable and flexible land use allocation model. Thirdly, it takes into account the factors of land use systems that have not been paid enough attention in the past, such as nitrogen emission, land use diversity and so on, which makes the model more perfect. Compared with other land use allocation optimization methods, the innovation points of this paper mainly include: (1) the economic development benefits and ecological service value benefits of the land use system are systematically analyzed and calculated; (2) the use of uncertain mathematical optimization model helps to obtain sustainable and flexible land use allocation; (3) comprehensive consideration of various factors. Under the constraints of economy, society, ecology, climate and technology, the land use planning scheme is obtained, which makes the model more perfect. Compared with uncertain mathematical models, most models mainly face two main problems in use: first, most models will face technical problems such as data obstacles in the application process, which makes the accuracy of model data inadequate. Second, due to the complexity of the actual situation and the variety of land use systems, it is difficult for most models to limit the uncertainty complexity in the real world for quantitative calculations. In addition, we believe that the uncertain mathematical model plays an important role in solving the problem of optimal allocation of land resources. This method can also be applied to other complex resource allocation systems and has been widely applied and improved.

However, this paper also has certain limitations. Based on the availability of data, this paper still does not fully consider the influencing factors and indicators of different land use configurations, the results obtained are limited and a large amount of data is needed to improve the accuracy of the research results [66]. In addition, the application of the model on larger scales, such as global, national and large-area, requires further research. In the future, we will consider combining the model in this paper with intelligent algorithmic models such as genetic algorithms, neural networks and particle swarm optimization, so as to carry out more scientific and effective land use optimization models [67].

**Author Contributions:** B.Q. and G.O. conceived and designed the research; B.Q. and G.O. collected, managed, Y.T. verified the data; M.Z., Y.Z. and H.M. calculated and analyzed the data and the results; S.L. and G.O. wrote the manuscript. All authors have read and agreed to the published version of the manuscript.

**Funding:** This research is supported by National Natural Science Foundation of China (project Nos. 42077432; 41401631); "the Fundamental Research Funds for the Central Universities", HUST: 2172021WKYXZD006.

**Institutional Review Board Statement:** Not applicable.

**Informed Consent Statement:** Not applicable.

**Data Availability Statement:** Not applicable.

**Acknowledgments:** The authors are grateful to the editors and anonymous reviewers for their insightful comments and suggestions.

**Conflicts of Interest:** The authors declare that they have no conflicts of interest.

## Nomenclature

Nomenclatures for parameters and variables

| | |
|---|---|
| $\pm$ | The interval value with lower and upper bounds |
| $\cong$ | Fuzzy equality |
| $\lesssim$ and $\gtrsim$ | Fuzzy inequality |
| $x$ | Decision variable |
| $i$ | $i$ represents the different district, where $i = 1$ for Jining, $i = 2$ for Zaozhuang, $i = 3$ for Heze and $i = 4$ for Ningyang |
| $j$ | $j$ means the type of land use, where $j = 1$ for agricultural land, $j = 2$ for construction land, $j = 3$ for green land, $j = 4$ for water land, $j = 5$ for other land, $j = 6$ for unused land; $t$ denotes time |
| $t$ | time period of land use planning, where $t = 1$ for 2021–2025, $t = 2$ for 2025–2030 |
| $E^{\pm}$ | Economic benefit of land use (RMB/km$^2$) |
| $ES^{\pm}$ | Ecological service benefit of land use (RMB/km$^2$) |
| $EP^{\pm}$ | Price of electricity (RMB/kWh) |
| $MUC^{\pm}$ | Unit maintenance cost per unit area of land use (RMB/km$^2$) |
| $WTUC^{\pm}$ | Cost of sewage treatment per unit area(RMB/km$^2$) |
| $STUC^{\pm}$ | Cost of solid waste disposal per unit area(RMB/km$^2$) |
| $WUC^{\pm}$ | Water consumption per unit area of land use (m$^3$/km$^2$) |
| $EUC^{\pm}$ | Electricity consumption per unit area of land use |
| $WP^{\pm}$ | Price of water (RMB/m$^3$) |
| $MGI^{\pm}$ | Maximum government investment in period $t$ (RMB) |
| $CUP^{\pm}$ | Grain production per unit area of agricultural land (ton/km$^2$); |
| $CD^{\pm}$ | Demand of grain (ton) |
| $AUP^{\pm}$ | Aquatic product output per unit area of water area (ton/km$^2$); |
| $AD^{\pm}$ | Demand of aquatic product (ton) |
| $TP^{\pm}$ | Total population (people) |
| $PCL^{\pm}$ | Maximum population per unit land area (people/km$^2$) |
| $AW^{\pm}$ | Supply of available water (ton) |
| $EW^{\pm}$ | Supply of electricity (kWh) |
| $LUA^{\pm}$ | Demand of labor force per unit area of agricultural land (people/km$^2$); |
| $ALA^{\pm}$ | Amount of available agricultural labor force (people) |
| $AM^{\pm}$ | Area of agricultural land (km$^2$) |
| $FR^{\pm}$ | Percentage of green land coverage (%) |
| $TA^{\pm}$ | Total land area (km$^2$); SHDI$_{i,t}$ means |
| $SHDI^{\pm}$ | Index of land use diversity at present |
| $WUD^{\pm}$ | Amount of waste water discharging amount per unite area(ton/km$^2$) |
| $AWC^{\pm}$ | Treatment capacity of waste water (ton) |
| $SUD^{\pm}$ | Discharge of solid waste per unite area of land use (ton/km$^2$) |
| $ASC^{\pm}$ | Handling capacity of solid waste discharge (ton) |
| $COD^{\pm}$ | Amount of COD discharge per unite area of land use (ton/km$^2$); |
| $DC^{\pm}$ | Maximum ammonia nitrogen capacity of environment (t) |
| $N^{\pm}$ | Amount of ammonia nitrogen discharge per unite area of land use (ton/km$^2$) |
| $DN^{\pm}$ | Maximum ammonia nitrogen capacity of environment (t) |
| $SO2^{\pm}$ | Amount of SO$_2$ discharge per unite area of land use (ton/km$^2$); |
| $DS^{\pm}$ | Maximum SO$_2$ capacity of environment (t) |
| $TA_t^{\pm}$ | Total area of study area (km$^2$) |

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
