# Peer review of "Optimal Modeling of Sustainable Land Use Planning under Uncertain at a Watershed Level: Interval Stochastic Fuzzy Linear Programming with Chance Constraints"

_land, doi:10.3390/land12051099_

Round 1
Reviewer 1 Report (Previous Reviewer 1)
good luck. can be accepted for publication.
Extensive editing of English language required
Author Response
Thank you very much for your comments. We have done our best to improve English language of this manuscript. Please see the revised version (land-2383375(V3)) in the attachment.

Reviewer 2 Report (Previous Reviewer 2)
Thank you for your addressing my comments carefully and I see that your revised paper is much improved. However, there are still a few typos, so make sure to correct them as much as you can throughout the work.
I have no further comment
Author Response
Thank you very much for your comments. We have done our best to reduce the spelling mistake and to improve English language of this manuscript. Please see the revised version (land-2383375(V3)) in the attachment.

This manuscript is a resubmission of an earlier submission. The following is a list of the peer review reports and author responses from that submission.
Round 1
Reviewer 1 Report
(1) The results of this paper are similar to their conclusions. Therefore, it is necessary to further summarize the innovation of the paper.
(2) Improve the quality of Table. 1. The unit economic benefit of land use
(3) Figure 2. Framework of UISFCL-LUP need more discussion, such as Constraint of land use system, in which water resources are in social system, please check.
(4) references of the last 5 years are required in this paper.
Author Response
RESPONSES TO THE REVIEWER 1’S COMMENTS
We are grateful to the Reviewer 1 for his/her insightful review. His/Her comments have contributed substantially to improving the paper. According to he/she, we have made efforts to significantly revise the manuscript, with the details explained as follows.
- The results of this paper are similar to their conclusions. Therefore, it is necessary to further summarize the innovation of the paper.
Response: Thank you for your comments.We have rearranged the findings and conclusions of the study and added an elaboration of the innovative aspects of the paper in the conclusion section to make the paper more innovative.
2.Improve the quality of Table. 1. The unit economic benefit of land use
Response: Thank you for your comments. We have modified the table of the article to a three-line table
3.Figure 2. Framework of UISFCL-LUP need more discussion, such as Constraint of land use system, in which water resources are in social system, please check.
Response: Thank you for your concern, which has been revised to place water resource in the environment section.
- references of the last 5 years are required in this paper.
Response: Thank you for your comments. We have added some references from the past five years according to the opinions and marked them in the text. Currently, this article has included a large number of references from the past five years.
Generally, we are deeply grateful to the reviewer 1 for his/her insight and careful review. His/her comments have greatly helped improve the paper. We also expressed our gratitude in the "Acknowledgment" section of the revised manuscript.
Reviewer 2 Report
Reviewer's comments:
I have some comments on your paper titled “Optimal modeling of sustainable land use planning under uncertain at a watershed level: Interval stochastic fuzzy linear programming with chance constraints”. Although the paper topic is interesting, and this may contribute to the literature on land management and development, many parts of the paper are still weak. Author(s) are highly suggested to address the following points to improve the work quality.
Specific comments:
1. Please further elaborate the Abstract section. Particularly, you should not provide all numbers of land types as presented.
2. Please further elaborate on the data section. You must provide a more detailed description of the data used in the model.
3. Please move the Figure 2 to before the first paragraph of Section 3. Please further elaborate this framework carefully.
4. Please further elaborate the model and its variables; please describe the variable selecting process
5. Please put the short definition of each variable right after the abbreviation (e.g., E (±) (i=1, j =1)) of the variable in Tables 1, 2, 3, rather than just the abbreviation of the variable. This would assist readers in better following/understanding the content of the tables provided.
6. The paper lacks the discussion section; please supplement this part in the updated version
7. The paper's limitation section is missing, so please address this in the next version.
Author Response
RESPONSES TO THE REVIEWER 2’S COMMENTS
We are grateful to the Reviewer 2 for his/her insightful review. His/Her comments have contributed substantially to improving the paper. According to he/she, we have made efforts to significantly revise the manuscript, with the details explained as follows.
- Please further elaborate the Abstract section. Particularly, you should not provide all numbers of land types as presented.
Response: Thank you for your comments. We have significantly revised the original version. The revised version has deleted the quantity of each land use type and further enriched the content. We believe that the research in this paper will enrich and improve the quantitative measurement and evaluation methods related to urban sprawl for the international academic community and presents a useful application of big data in identifying urban sprawl. A more microscopic, newly added outlying expansion patch scale for detecting the location and degree of a city’s spread will effectively replace using the entire city as a unit to measure urban sprawl.
- Please further elaborate on the data section. You must provide a more detailed description of the data used in the model.
Response: Thank you for your comments. The revised version has added relevant text for the data source section.
- Please move the Figure 2 to before the first paragraph of Section 3. Please further elaborate this framework carefully.
Response: Thank you for your comments.Figure 2 has been moved to precede the first paragraph of section 3 and further elaborates this framework.
- Please further elaborate the model and its variables; please describe the variable selecting process
Response: Thank you for your comments.The model and variables are further elaborated in this paper: textual explanations are added to the decision variables section and a table has been recreated in the constraints section to explain them. In addition, the justification for the choice of variables is added.
- Please put the short definition of each variable right after the abbreviation (e.g., E (±) (i=1, j=1)) of the variable in Tables 1, 2, 3, rather than just the abbreviation of the variable. This would assist readers in better following/understanding the content of the tables provided.
Response: Thanks for your advice, but due to the page size limit, we cannot put a short definition of the variable in the original table, so we detail each parameter in the attachment section.
- The paper lacks the discussion section; please supplement this part in the updated version
Response: Thank you for your comments.We have reorganized the study findings, revised the findings section to Conclusions and Discussions, and added discussion-related content to make the article more structured.
- The paper's limitation section is missing, so please address this in the next version.
Response: Thank you for your comments.We have added a new limitations section to the article in the newly revised version.
Generally, we are deeply grateful to the reviewer 2 for his/her insight and careful review. His/her comments have greatly helped improve the paper. We also expressed our gratitude in the "Acknowledgment" section of the revised manuscript.

Reviewer 3 Report
I would like to thank the author for the effort made in the paper. The scientific contribution of the paper is high. The approach is perfectly applied and all constraints were taken into consideration. However, I believe the paper could be more enhanced with these minor corrections:
- The abstract of the paper describes well the scoop of the paper. But, there is too much results especially for land use results. It is best to modify that section with more meaningful interpretation given from these results.
- The introduction is a little long but it is good. In the lines 102-115, you referred that it was based on the previous scholar studies in the upper section. But are the drawbacks of mathematical models have been interpreted by the authors or they were extracted from research studies? please add some references for each shortcoming.
-In the lines 160-170: The advantages of the model in the introduction should be moved to the discussion section and there where you should discuss widely the advantages/the performance and drawbacks/or lacks of accuracy in the Model.
figure 1 must be enhanced: The figure is too cluttered, I suggest the following corrections:
1- histograms are not legible and cannot be well interpreted and correlated with the legend. So it is suitable to remove them from the map and represent these results in a separate table.
2- In the first two maps, try to add some referenced labels for some areas, so foreign readers could understand the study area location. it is pointless to fragment the area without labelling them(or some of them).
3- The labels you put are covering the map and land use types. Try to change the background of the labels to none (or "hollow" if you are working with ArcGIS).
- Title and table 1 are present in two separated pages. please correct
- Sections 2 "constraints": I think all the cited section will be more suitable in a table (constraint name, definition, equation). Add a paragraph to speak about the contribution of these constraints, input for models, etc.
- Figure 3 : please change the pattern/or the colour of the histograms, it is difficult to differentiate between the land use types.
- In the result section, you could summarize the results of different land use allocations in a table.
- When reading the result section and getting to achieve the whole paper. I am starting to believe that the paper have more methodological trend and simple application than scientific approach. That's why the results of different land use allocations should be more discussed. The accuracy of the model have not also been discussed. Have you found any other studies that you could correlate your work with ? could you explain also the land use allocation results ? Could you please discuss the performance of the model, the drawbacks, how can we enhance the accuracy of the model, etc ?
- The conclusion is good but you could add a section of perspective.
Author Response
RESPONSES TO THE REVIEWER 3’S COMMENTS
We are grateful to the Reviewer 3 for his/her insightful review. His/Her comments have contributed substantially to improving the paper. According to he/she, we have made efforts to significantly revise the manuscript, with the details explained as follows.
1.The abstract of the paper describes well the scoop of the paper. But, there is too much results especially for land use results. It is best to modify that section with more meaningful interpretation given from these results.
Response: Thank you for your comments. The revised version has removed the quantitative results of land use types and added content related to the actual application results of the model.
2.The introduction is a little long but it is good. In the lines 102-115, you referred that it was based on the previous scholar studies in the upper section. But are the drawbacks of mathematical models have been interpreted by the authors or they were extracted from research studies? please add some references for each shortcoming.
Response: Thank you for your comments. This content is based on the author's review and summary of papers on the application of relevant models in the academic community, as well as the shortcomings and problems found in the actual application of relevant models in academic research.
3.In the lines 160-170: The advantages of the model in the introduction should be moved to the discussion section and there where you should discuss widely the advantages/the performance and drawbacks/or lacks of accuracy in the Model.
Response: Thank you for your comments.We have moved the model advantages section in introduction to the discussion section and elaborated the advantages and disadvantages of the model in the discussion section.
4.figure 1 must be enhanced: The figure is too cluttered, I suggest the following corrections:
4.1- histograms are not legible and cannot be well interpreted and correlated with the legend. So it is suitable to remove them from the map and represent these results in a separate table.
Response: Thank you for your comments. The modified version has removed the histogram and displayed the relevant histogram data in a separate list for the original version。
4.2- In the first two maps, try to add some referenced labels for some areas, so foreign readers could understand the study area location. it is pointless to fragment the area without labelling them(or some of them).
Response: Thank you for your comments. The revised version has added corresponding geographic labels.
4.3- The labels you put are covering the map and land use types. Try to change the background of the labels to none (or "hollow" if you are working with ArcGIS).
Response: Thank you for your comments. The revised version has changed the background of the labels to none.
5.Title and table 1 are present in two separated pages. please correct
Response: Thank you for your concern, we have placed the title of Form 1 and the table content on one page.
6.Sections 2 "constraints": I think all the cited section will be more suitable in a table (constraint name, definition, equation). Add a paragraph to speak about the contribution of these constraints, input for models, etc.
Response: Thank you for your comments.We have placed the constraints in a table and further elaborated on the model and its variables.
7.Figure 3 : please change the pattern/or the colour of the histograms, it is difficult to differentiate between the land use types.
Response: Thank you for your comments.We have redrawn the bar chart as you suggested, but due to the complexity of the results and the size of the data, it is still not possible to present the land use types in a clearer way, so in order to be consistent with the graphs of the whole paper, we have chosen to use this type of bar chart, but enlarged it to make the content of the chart more clearly distinguishable.
8.In the result section, you could summarize the results of different land use allocations in a table.
Response: Thank you for your comments.We have made a table to summarize the results of the different land use allocations as you suggested, but we found that the table presentation was not aesthetically pleasing and that the data results were too dense to visualize the findings of the paper, so we have continued with the original bar chart format.
9.When reading the result section and getting to achieve the whole paper. I am starting to believe that the paper have more methodological trend and simple application than scientific approach. That's why the results of different land use allocations should be more discussed. The accuracy of the model have not also been discussed. Have you found any other studies that you could correlate your work with ? could you explain also the land use allocation results ? Could you please discuss the performance of the model, the drawbacks, how can we enhance the accuracy of the model, etc ?
Response: Thank you for your comments.We have added a discussion of the accuracy of the model, including its strengths and weaknesses and future directions and optimization ideas, as well as a discussion in the conclusion section that provides a more objective evaluation of the model used.
10.The conclusion is good but you could add a section of perspective.
Response: Thank you for your comments.We have reworked the conclusion section, added new discussion, and improved the article.
Generally, we are deeply grateful to the reviewer 3 for his/her insight and careful review. His/her comments have greatly helped improve the paper. We also expressed our gratitude in the "Acknowledgment" section of the revised manuscript.
